# Van der Waals Epitaxial Growth of ZnO Films on Mica Substrates in Low-Temperature Aqueous Solution

**Hou-Guang Chen *** , **Yung-Hui Shih, Huei-Sen Wang *, Sheng-Rui Jian** , **Tzu-Yi Yang and Shu-Chien Chuang**

Department of Materials Science and Engineering, I-Shou University, Kaohsiung City 84001, Taiwan; yhshi@isu.edu.tw (Y.-H.S.); srjian@isu.edu.tw (S.-R.J.); ziyiiyun119@gmail.com (T.-Y.Y.); r262524526@gmail.com (S.-C.C.)
* Correspondence: houguang@isu.edu.tw (H.-G.C.); huei@isu.edu.tw (H.-S.W.)

**Abstract:** In this article, we demonstrate the van der Waals (vdW) epitaxial growth of ZnO layers on mica substrates through a low-temperature hydrothermal process. The thermal pretreatment of mica substrates prior to the hydrothermal growth of ZnO is essential for growing ZnO crystals in epitaxy with the mica substrates. The addition of sodium citrate into the growth solution significantly promotes the growth of ZnO crystallites in a lateral direction to achieve fully coalesced, continuous ZnO epitaxial layers. As confirmed through transmission electron microscopy, the epitaxial paradigm of the ZnO layer on the mica substrate was regarded as an incommensurate van der Waals epitaxy. Furthermore, through the association of the Mist-CVD process, the high-density and uniform distribution of ZnO seeds preferentially occurred on mica substrates, leading to greatly improving the epitaxial qualities of the hydrothermally grown ZnO layers and obtaining flat surface morphologies. The electrical and optoelectrical properties of the vdW epitaxial ZnO layer grown on mica substrates were comparable with those grown on sapphire substrates through conventional solution-based epitaxy techniques.

**Keywords:** zinc oxide; van der Waals epitaxy; hydrothermal growth; Mist-CVD; mica





## 1. Introduction

Zinc oxide (ZnO) has attracted intensive research efforts for its versatile applications in transparent electronics, solar cells, gas sensors, light-emitting diodes, laser diodes, and photodetectors due to its excellent optoelectronic properties including wide band gap (3.37 eV), high exciton binding energy at room temperature (60 meV), and high optical transparency within the visible spectrum [1–8]. The epitaxial growth of ZnO layers is essential for the development of advanced ZnO-based optoelectronic devices. For the conventional epitaxy paradigm, due to the strong covalent interactions at the heterointerface, the heteroepitaxial growth of semiconductors or oxides on single-crystal substrates requires a lattice-matched or small lattice-mismatched system, leading to only limited combinations of materials suitable for heteroepitaxial growth. Recently, the van der Waals (vdW) epitaxy, regarded as an incommensurate epitaxy, has been considered to enable an heteroepitaxy with large lattice mismatching [9,10]. In contrast to a conventional covalent epitaxy, the van der Waals epitaxy, mediated by the weak van der Waals force, can overcome the limitation of the lattice mismatch between the overlayer and substrate. To date, a considerable number of experimental and computational studies have been made on the van der Waals epitaxy, such as the epitaxial growth of the two-dimensional layered (2D) materials on either 2D or 3D (e.g., sapphire) single-crystal substrates [11–14]. In addition, the epitaxial growth of 3D materials on van der Waals crystals whose surface is chemically inert, and the lack of dangling bonds, such as muscovite mica, is also considered a category of the vdW epitaxy paradigm [15]. By taking advantage of the weak-interlayer van der Waals force, the ultra-thin mica sheet, with a thickness in the range of several micrometers,

can be readily obtained by cleaving along the (001) plane; the resulting muscovite mica sheet is highly transparent and flexible, making it a suitable substrate for application in flexible optoelectronics [15–17]. Thus, to realize an advanced functional oxide heteroepitaxy, there is considerable interest in the vdW epitaxial growth of various functional-oxide nanostructures or layers on the muscovite mica substrates [16–22].

In the last few years, many articles have been devoted to the study of the vdW epitaxial growth of ZnO nanostructures or layers on muscovite mica substrates by using the vapor-phase deposition process [23]. The Al-doped ZnO epitaxial layers, grown on mica substrates, exhibited good flexibility and superior durability [17]. Li et al. reported the epitaxial growth of quasi-2D ZnO single crystal plates on mica substrates by applying the technique of pulse-laser-deposition (PLD)-assisted vdW epitaxy. Moreover, due to weak interfacial interaction, the resulting ZnO plates could be easily transferred from the mica substrate onto the $SiO_2/Si$ substrate to demonstrate the applications of vdW epitaxial ZnO crystals in optoelectronic devices, including the self-powered ultraviolet (UV) photodetector and UV light-emitting diode, respectively [24]. Thus far, the vdW epitaxial growth of ZnO films on mica substrates has been mainly accomplished through the high-vacuum vapor phase epitaxy technique. Although the solution-phase vdW epitaxy of ZnO nanowires on mica substrates has been proven feasible [25], little attention has been given to the point of vdW epitaxial growth of ZnO layers on mica substrates by the solution-based or atmospheric-pressure process. In this study, we employed hydrothermal growth and atmospheric pressure solution-processed mist chemical vapor deposition (Mist-CVD) to implement the vdW epitaxial growth of ZnO films on mica substrates, because of their low-cost equipment and non-vacuum system [26]. Furthermore, the study also proposed a seed-assisted hydrothermal growth of vdW epitaxial ZnO films by combining the Mist-CVD process with low-temperature hydrothermal growth, and investigated the effect of growth conditions on the vdW epitaxial growth of ZnO films on mica substrates.

## 2. Materials and Methods

A freshly cleaved muscovite mica (V-1 grade) was used as the substrate for the van der Waals epitaxial growth of ZnO crystals. For hydrothermal growth of ZnO, the samples were immersed in an aqueous solution of 0.1 M zinc nitrate ($Zn(NO_3)_2 \cdot 6H_2O$, 99%, Showa) and 0.1 M hexamethylenetetramine (HMT) (($CH_2)_6N_4$, 99%, Showa) at 90 °C for 5 h. A different concentration of tri-sodium citrate dihydrate ($Na_3C_6H_5O_7 \cdot 2H_2O$, 99%, Aencore) was added to the solution. The surface pretreatment of mica substrates was performed by thermal annealing in air at various temperatures for 3 h. For seed-assisted hydrothermal growth, the atmospheric-pressure Mist-CVD process was employed to implement the van der Waals epitaxial growth of ZnO seeds on mica substrates, followed by subsequent hydrothermal growth of ZnO layers. The configuration of the Mist-CVD apparatus used in this work is illustrated in Figure 1. The apparatus consisted of a mist generator and a horizontal quartz-tube furnace. In this study, the zinc source was zinc acetic dehydrate ($Zn(CH_3COO)_2 \cdot 2H_2O$, 99%, Showa). An amount of 0.1 M zinc acetic was dissolved in a solution mixture of deionized water and acetic acid (70:30). The precursor solution was atomized into liquid aerosol particles by a 2.4 MHz ultrasonic transducer, and the aerosols formed were transferred into a tube furnace with the carrier gas of either nitrogen ($N_2$, 99.99% purity) or oxygen ($O_2$, 99.99% purity), respectively, at a flow rate of 2 L/min. The temperature and growth time for the vdW epitaxial growth of ZnO seeds on mica were 600 °C and 30 min, respectively.

To start with, surface morphologies of the samples were observed by field-emission scanning electron microscopy (SEM, Hitachi, S4700). The structural properties of ZnO films grown on mica were examined using an X-ray diffractometer (XRD, PANalytical, X'Pert PRO-MPD). A cross-sectional sample was prepared with the focused ion beam technique (FIB, FEI, Nova-200 NanoLab). The ZnO–mica interface was also observed by high-resolution transmission electron microscopy (HRTEM, JEOL, JEM-2100F). The HRTEM image of the ZnO–mica interface was further analyzed by Gatan Digital Micrograph[TM]

software. The geometric phase analysis (GPA), developed by Hüytch et al., was performed on the HRTEM image to produce the strain map around the interface [27]. The Strain++ program was used for GPA [28–30]. Next, the room-temperature photoluminescence (PL) spectrum was recorded by a Horiba Jobin Yvon HR800 system, with a 325 nm He-Cd laser as an excitation source. The transmission spectrum was obtained using a UV-Vis spectrophotometer (UV-Vis, Thermal Scientific, Evolution 201). Finally, the electrical resistivity, carrier concentration, and mobility of the ZnO films were measured by a Hall-effect measurement analyzer (MarChannel, AHM-800B).

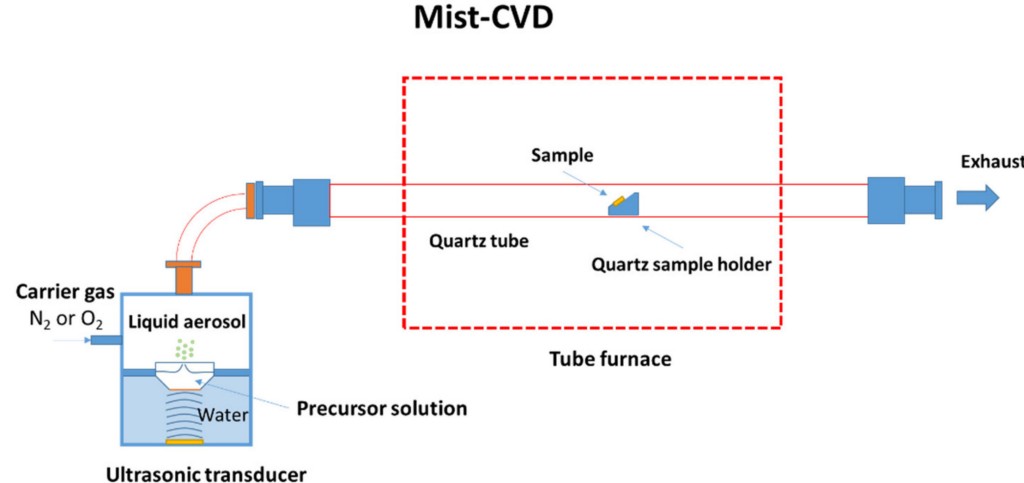

**Figure 1.** The schematic illustration of Mist-CVD system.

## 3. Results

### 3.1. Van der Waals Epitaxial Growth of ZnO Films on Mica Substrates by Low-Temperature Hydrothermal Growth Process

Figure 2a demonstrates the SEM image of ZnO micro-rods hydrothermally grown on the as-cleaved mica substrate with 0.25 mM sodium citrate. The image shows that the majority of ZnO micro-rods were oriented in random directions on the mica substrates. Despite a few ZnO micro-rods with in-plane alignment existing on the sample (Figure 2b), the epitaxial growth of ZnO crystals on as-cleaved mica, without other pretreatments in the low-temperature aqueous solution, was indeed very difficult. Although a freshly cleaved mica surface was usually used as substrates for the van der Waals epitaxy of various oxide systems, the mica surface pretreatment was also the critical for the van der Waals epitaxy of other semiconductor materials [31,32]. Thermal annealing is a common means to modify the substrate surface for epitaxial growth [33,34]. Hence, we performed the thermal annealing treatment of the mica substrates at various temperatures. Figure 2c shows the hydrothermal growth of ZnO micro-rods performed on the 400 °C annealed mica substrate; notably, several epitaxial ZnO micro-rods, exhibiting in-plane alignment, can be observed on such substrates. Furthermore, with the increase in annealing temperature to 600 °C, a high density of vertical aligned ZnO micro-rods grew on mica substrates (Figure 2d). The magnified SEM image (Figure 2e) shows an in-plane alignment of most of the vertically aligned ZnO micro-rods, indicating that the high-temperature annealing treatment (~600 °C) of mica substrates is beneficial to the van der Waals epitaxial growth of ZnO on mica in the aqueous solution at low temperature (~90 °C). In addition, highly dense in-plane aligned ZnO rods, partially coalescing into continuous layers, can be found in the other regions of the same sample, as shown in Figure 2f. For conventional hydrothermal growth of ZnO micro-rods, the addition of tri-sodium citrate into the growth solution can inhibit the growth along the *c*-axis of ZnO and promote the growth of ZnO along the lateral direction. To demonstrate the addition effect of tri-sodium citrate in the zinc nitrate/HMT precursor solution for the growth of ZnO crystals on mica substrates, the hydrothermal

growth of ZnO crystals in the solutions containing various citrate concentrations was implemented on 600 °C annealed mica substrates. Figure 3a shows the SEM image of the ZnO micro-rods epitaxially grown on mica substrates in the precursor solution without citrate. The diameters of the ZnO micro-rods were, obviously, much smaller (~1.3 µm) than those grown (3 µm) in the solution with 0.25 mM citrate (Figure 2d). Remarkably, a fully continuous ZnO layer was formed on the mica substrate in the solution containing the citrate concentration of 0.5 mM, as shown in Figure 3b, indicating that the addition of tri-sodium citrate in the precursor solution is a key factor for growing fully coalesced ZnO films on mica substrates.

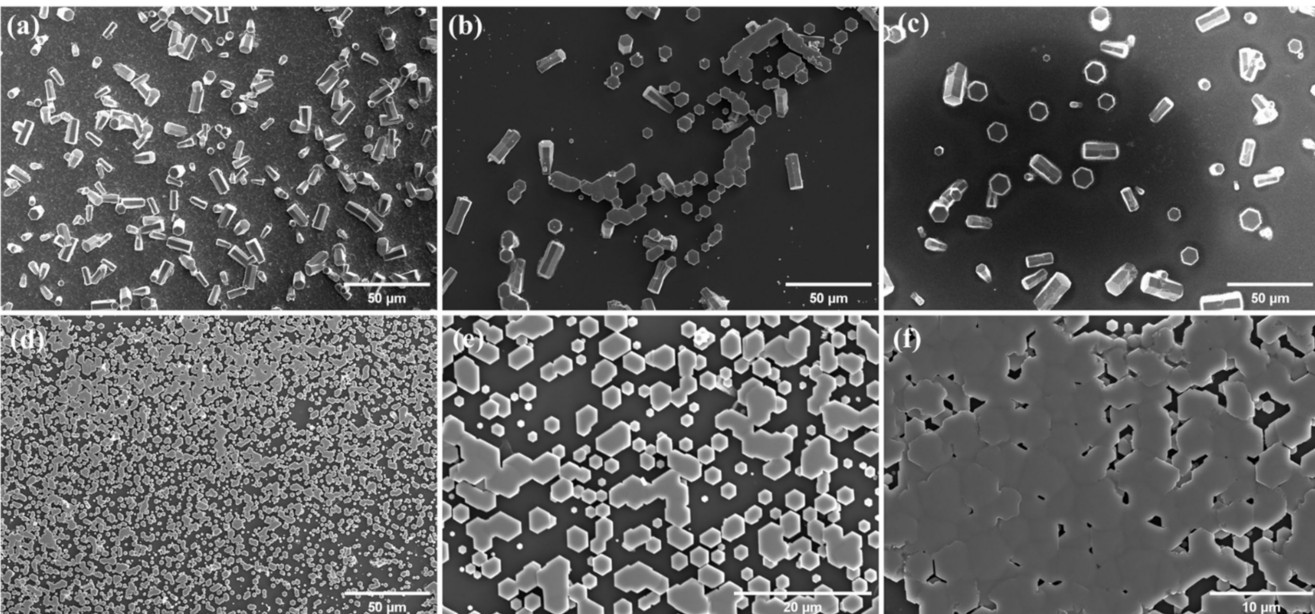

**Figure 2.** (**a**) SEM images of randomly oriented hydrothermally grown ZnO micro-rods, from the solution containing 0.25 mM sodium citrate, on as-cleaved mica substrate and (**b**) a few ZnO micro-rods with in-plane alignment existing on the other region of the same sample. SEM images of ZnO micro-rods hydrothermally grown on (**c**) 400 °C and (**d**) 600 °C annealed mica substrates. (**e**) Magnified SEM image of (**d**). (**f**) Partially coalesced ZnO layers formed on the other region of 600 °C annealed mica substrate.

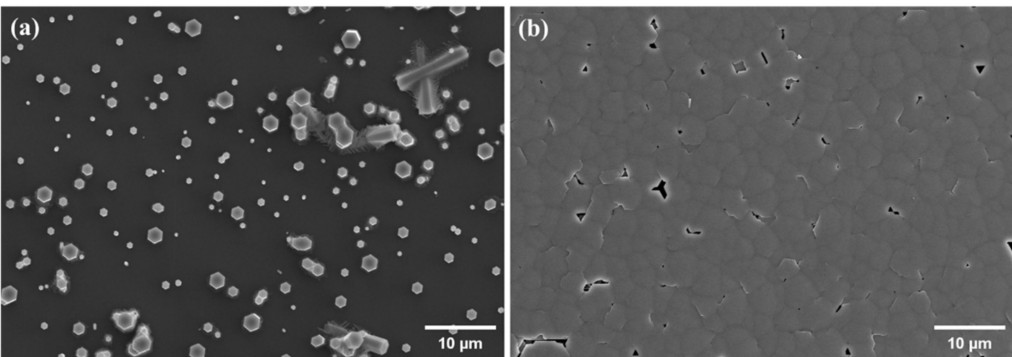

**Figure 3.** SEM images of ZnO micro-rods hydrothermally grown on 600 °C annealed mica substrates from the solution containing (**a**) 0 mM and (**b**) 0.5 mM sodium citrate.

The XRD measurement was carried out to gain insight into the microstructural properties of the epitaxial ZnO layer grown on the mica substrate in the solution with 0.5 mM citrate. In Figure 4a, the profile of the θ-2θ scan of the continuous ZnO film grown on the mica substrate revealed the preferred orientation of the *c*-axis, consisting of the re-

sults of SEM observation, as shown in Figure 3b. Figure 4b displays the XRD φ-scan profiles of ZnO {10$\bar{1}$1} and mica {202} planes. Based on the results of XRD φ-scans, the ZnO film was in epitaxy with the mica substrate, with the in-plane orientation relationship of (0001)$_{ZnO}$||(001)$_{mica}$ and [11$\bar{2}$0]$_{ZnO}$||[10]$_{mica}$. To evaluate the mosaic spread and epitaxial quality of the epitaxial ZnO layer, hydrothermally grown on the mica substrate, we recorded the X-ray rocking curve (XRC) profile of the ZnO (0002) plane, as shown in Figure 4c. The full-width at half-maximum (FWHM) value of XRC, for the (0002) plane of ZnO, extracted by fitting a single pseudo-Voigt function was 0.51°, which was significantly broader than those of ZnO epitaxial layers grown on sapphire substrates under the similar solution conditions [35,36].

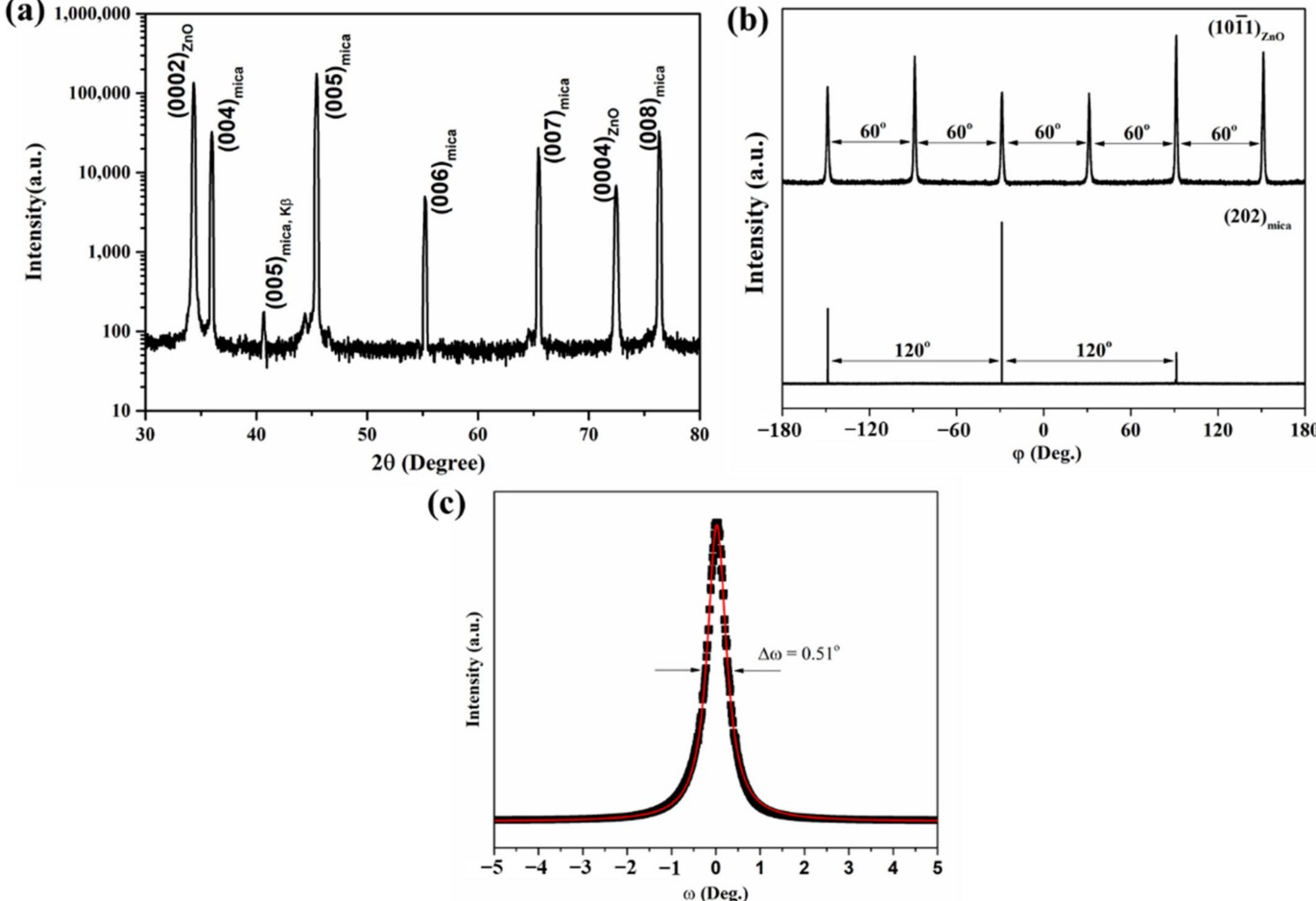

**Figure 4.** (**a**) XRD θ-2θ profile, (**b**) XRD φ-scan of the ZnO {10$\bar{1}$1} and mica {202} reflections, and (**c**) Rocking curve profile of the ZnO (0002) reflection for the continuous ZnO film hydrothermally grown on 600 °C annealed mica substrate from a solution containing 0.5 mM sodium citrate.

We explored the interfacial characteristics of the vdW epitaxial ZnO micro-rods and film hydrothermally grown on mica substrates by cross-sectional transmission electron microscopy. Figure 5a,c show the cross-sectional TEM bright-field images of the ZnO micro-rods and film hydrothermally grown on the 600 °C annealed mica substrates from the solutions containing 0.25 mM and 0.5 mM sodium citrate, respectively. The corresponding selective-area-electron-diffraction (SAED) patterns were taken from their interfacial regions, as shown in Figure 5b,d, respectively, showing that the orientational relationship between ZnO and mica was (0001)$_{ZnO}$||(001)$_{mica}$ and [11$\bar{2}$0]$_{ZnO}$||[310]$_{mica}$ for the ZnO micro-rods grown from solution with 0.25 mM sodium citrate (Figure 5b) and (0001)$_{ZnO}$||(001)$_{mica}$ and [11$\bar{2}$0]$_{ZnO}$||[10]$_{mica}$ for the ZnO film grown from the solution with 0.5 mM sodium

citrate (Figure 5d). Due to the (001) plane of mica possessing a quasi-hexagonal symmetry, the angle between directions [10] and [310] was 59.96°, which is very close to 60°. Therefore, while the ZnO <11$\bar{2}$0> directions were parallel to the mica <010> directions, the other ZnO <11$\bar{2}$0> directions were almost nearly parallel to mica <310> directions. Based on the SAED patterns, the epitaxial relationship between the ZnO layer and mica substrate can be estimated as $(0001)_{ZnO} || (001)_{mica}$ and $[11\bar{2}0]_{ZnO} || [010]_{mica}$, in agreement with the results of the XRD φ-scan. Based on the cross-sectional TEM images, the height of the resulting ZnO micro-rods (~4.2 μm) grown from the solution with 0.25 mM sodium citrate was higher than that from the solution with 0.5 mM sodium citrate (~2.3 μm); hence, the higher concentration of sodium citrate obviously reduced the growth rate along the *c*-axis of ZnO crystals, consistent with the results of other reported studies in the literature. The effect of sodium citrate as a crystal habit modifier on the morphology of ZnO has been widely studied in the previous works [37–40].

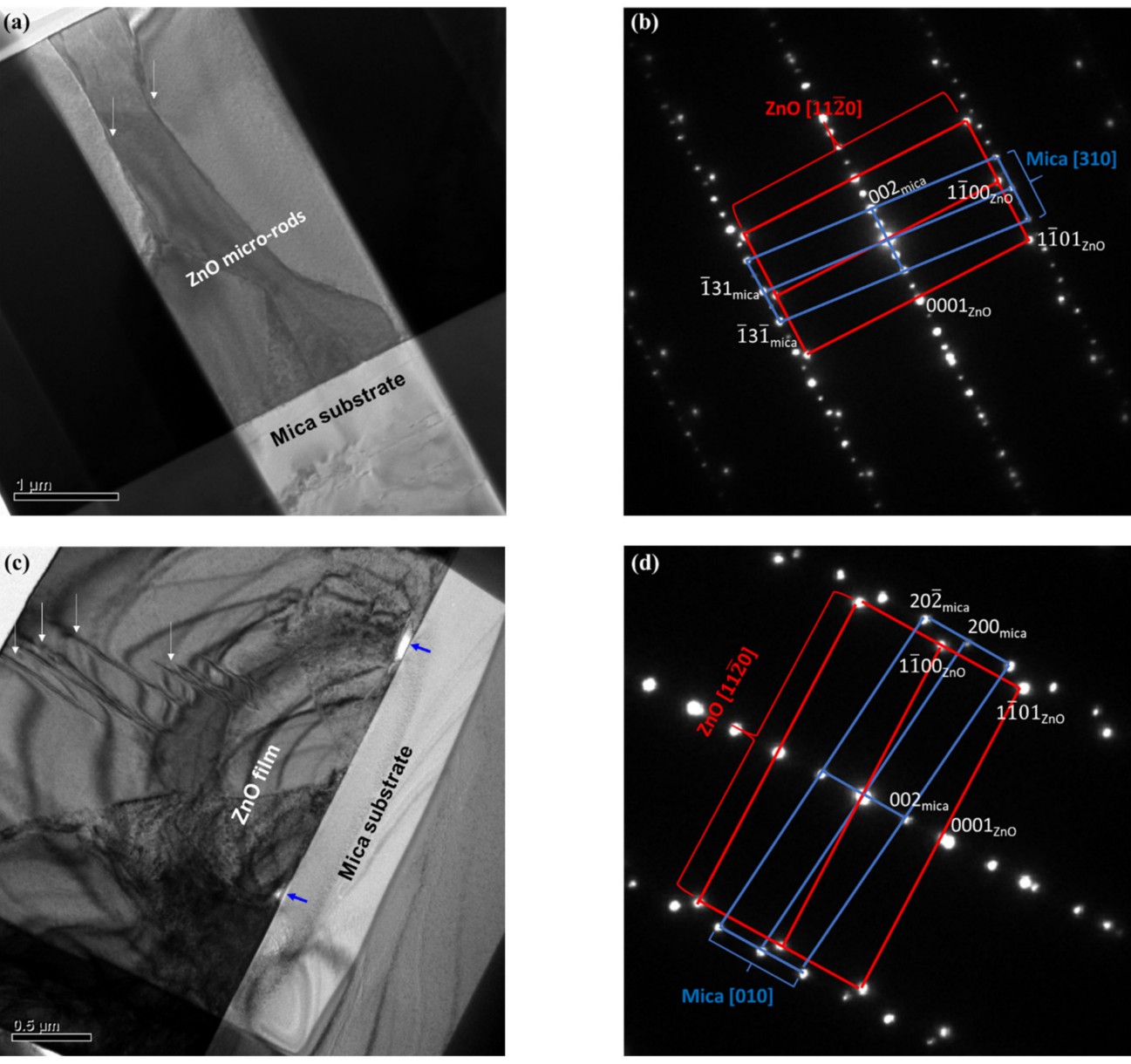

**Figure 5.** (**a**) Cross-sectional bright-field TEM images of ZnO micro-rods and film hydrothermally grown on mica substrates from solutions containing (**a**) 0.25 mM and (**c**) 0.5 mM sodium citrate. (**b**,**d**) Corresponding SAED patterns recorded at the interfacial regions.

In the previous report of the solution-phase vdW epitaxy of ZnO wires on the mica substrate, the initial growth behavior of ZnO on mica, following the three-dimensional island growth, was named the Volmer–Weber (VW) model. Two or more tiny wires grown from nucleation islands were merged to form a single ZnO wire with a larger diameter during the growing process. However, despite the coalesced ZnO wire being confirmed as a single crystalline, the resulting wires would suffer certain lattice distortions or strains due to the imperfect attachment of the different adjacent wires during the growth process [25]. In this work, remarkably, the cross-sectional TEM image revealed the threading dislocations present in either the micro-rods or the epitaxial layer, as indicated by white arrows in Figure 5a,c. It is considered that the threading dislocations formed at the merging boundaries accommodated the misorientations, originated from the imperfect attachment among adjacent crystallites, and resulted in the broadening of the XRC peak (also see Figure 4c) [41]. Furthermore, because of the less surface migration of growth species at low growth temperature, the spaces, among these nucleation islands, persisted during the growth process and left several tiny voids, as marked by blue arrows in Figure 5c, at the heterointerface between the ZnO film and mica substrate [25].

The interfacial structure between the ZnO film and mica substrate (Figure 5c) was visualized by using high-resolution transmission electron microscopy (HRTEM), viewed along the mica [10] zone-axis, as shown in Figure 6a. An atomically abrupt interface, without an obvious intermediate layer, can be observed at the ZnO/mica interface region. The corresponding fast Fourier transform (FFT) pattern (Figure 6b) of the HRTEM image was consistent with the SAED pattern (Figure 5d). The Fourier-filtered image reconstructed by utilizing ZnO $1\bar{1}00$ and mica 200 reflections is shown in Figure 6c. The measured average lattice spacings of $(1\bar{1}00)_{ZnO}$ and $(200)_{mica}$ were about 0.2814 nm and 0.260 nm, respectively; the values were nearly identical to the values of the standard JCPDS for ZnO (JCPDS card No. 36-1451) and muscovite mica (JCPDS card No. 06-0263), respectively. At the interface, the periodic misfit dislocations (indicated by red arrows) were evident in the Fourier-filtered image (Figure 6c), with every 12 planes of ZnO $(1\bar{1}00)$ matching with 13 planes of mica (200), which is the same as the results reported by Utama et al. [23]. The calculated lattice mismatch between $(1\bar{1}00)$ZnO and (200)mica was about 8.23%, which was accommodated by matching $12 \times d(1\bar{1}00)$ZnO with $13 \times d(200)$mica. Due to the near-strain relaxation of ZnO, the incommensurate van der Waals epitaxy was considered as the valid paradigm for epitaxial growth of the ZnO nanowire on the mica substrate.

For conventional covalent heteroepitaxy, the epitaxial films are strained to match the underlying substrate; hence, the coherently strained layers exist adjacent to the interface. To explore the lattice strain effect in the epitaxial ZnO layer, geometric phase analysis (GPA) based on the HRTEM image was conducted to produce nanoscale strain maps. Figure 7a shows the other HRTEM image of the ZnO layer/mica substrate interface, viewed along ZnO $[11\bar{2}0]$ direction. The $x$- and $y$-axis were parallel to the ZnO $[1\bar{1}00]$ and ZnO [1] directions, respectively. The $1\bar{1}00$ and 0001 reflections of the ZnO were chosen from the corresponding FFT pattern to implement geometric phase analysis (Figure 7b). The GPA strain maps of strain components $\varepsilon_{xx}$ and $\varepsilon_{yy}$ obtained from the region adjacent to the interface are illustrated in Figure 7c,d, respectively; notably, both strain maps of $\varepsilon_{xx}$ and $\varepsilon_{yy}$ exhibited a nearly uniform distribution of the strain with the value close to zero ($0.0 \pm 0.01$) in the whole region of ZnO. The absence of a gradual variation in strain along the growth direction indicated that the strained layer was not present around the interface. Therefore, the epitaxial growth of ZnO on mica was nearly fully relaxed. In this work, the epitaxial growth paradigm of ZnO film on the mica substrate was regarded as an incommensurate van der Waals epitaxy.

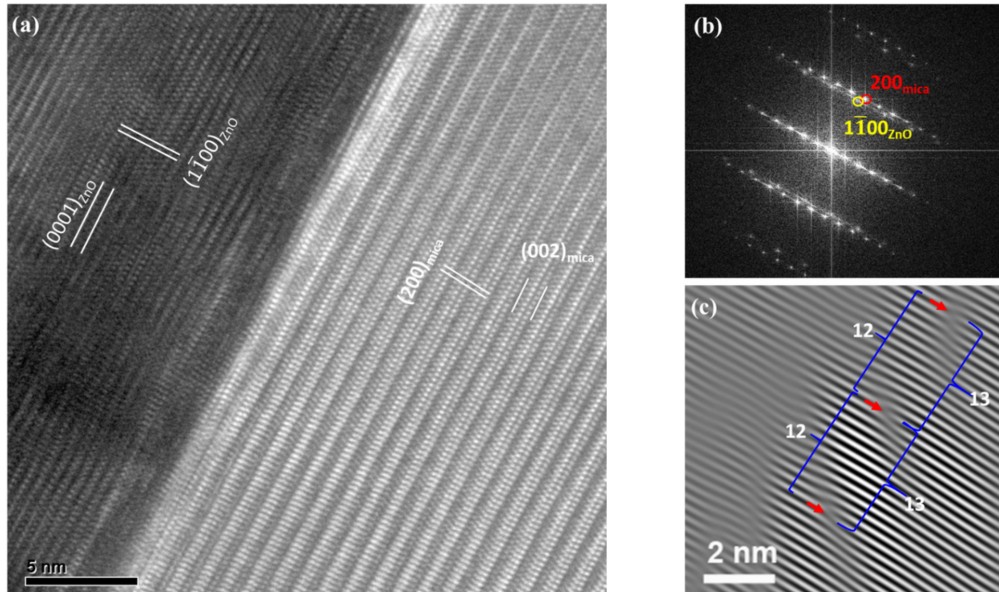

**Figure 6.** (**a**) HRTEM image of the ZnO layer–mica substrate interface viewed along the mica [10] zone axis. (**b**) Corresponding FFT pattern of (**a**). (**c**) Fourier-filtered image reconstructed utilizing ZnO $1\bar{1}00$ and mica 200 reflections. The black triangles in the Fourier-filtered image indicate the position of misfit dislocation.

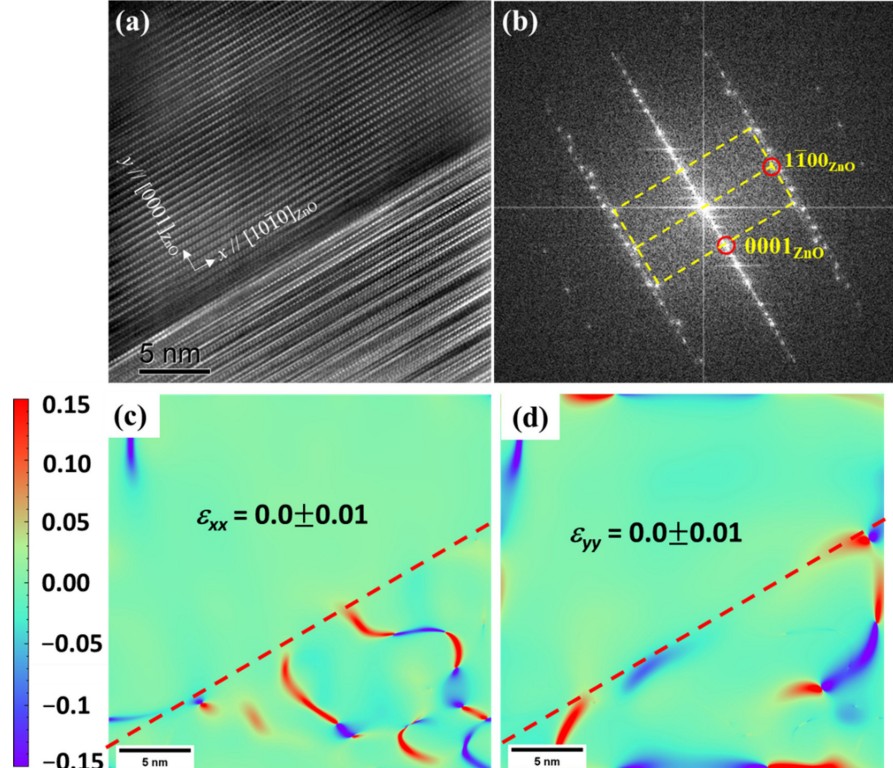

**Figure 7.** (**a**) HRTEM image of the region adjacent to the interface between the ZnO layer and the mica substrate viewed along the ZnO [$11\bar{2}0$] zone axis. (**b**) Corresponding FFT pattern of (**a**). The $1\bar{1}00$ and 0001 reflections of the ZnO were chosen to implement geometric phase analysis. The GPA strain maps of strain components (**c**) $\varepsilon_{xx}$ and (**d**) $\varepsilon_{yy}$ obtained from the region adjacent to the interface. The red-dashed lines indicate the interface.

*3.2. Van der Waals Epitaxial Growth of ZnO Films on Mica Substrates through the High-Temperature Seed-Layer-Assisted Hydrothermal Growth*

For solution-phase van der Waals epitaxial growth of ZnO films on mica, less surface migration of Zn-related species in the low-temperature environment led to the growth of ZnO islands with a sparse nucleation density at the initial growth stage. As a result, the coalesced epitaxial layer, following hydrothermal growth, featured a mosaic morphology with large misorientations. Namely, the increase in nucleation density of ZnO crystallites on mica at the initial growth stage is a straightforward strategy to ameliorate the growth quality of ZnO epitaxial layers on mica. In this study, the high-temperature seed-layer-assisted hydrothermal growth was performed. The seed layers can act as nucleation sites to facilitate the formation of epitaxial ZnO crystallites on mica substrates with high density. Prior to the hydrothermal growth of ZnO layers, the epitaxial ZnO seeds were grown on the mica substrate using an atmospheric-pressure Mist-CVD process for 30 min. Considering that the Mist-CVD process was carried out at high temperature (600 °C) under atmospheric pressure, the as-cleaved mica substrates without annealing treatments were used as the substrates for the vdW epitaxial growth of ZnO seeds. Taking advantage of the elevated temperature, the adatom received sufficient thermal energy to enhance surface migration, promoting the ZnO crystallite nucleation on mica at the initial growth stage and eliminating the defects or/and imperfect arrangement in lattices. In order to scrutinize the effect of ambient conditions on the growth behavior of ZnO seeds on mica during the Mist-CVD process, the nitrogen ($N_2$) and oxygen ($O_2$) were used as the carrier gases for the growth of ZnO seed layers on mica substrates, designated as Type A and Type B seeds, respectively. Figure 8a,b show the SEM images of two types of ZnO seeds grown on mica substrates. The surface morphologies revealed most of the ZnO crystallites with hexagonal facets and an in-plane alignment on both samples, and it suggested that the ZnO seeds were epitaxially grown on mica substrates by the Mist-CVD process under either nitrogen or oxygen ambient conditions. Two forms of ZnO crystallites were observed with different morphologies grown on the mica substrate under nitrogen ambient (Type A): One of the ZnO crystallites presented a hexagonal rod-like morphology with a sharp tip, and an average diameter of about 1.5 μm; the other form exhibited a flat top morphology with a larger diameter of 3~5 μm. Notably, several ZnO crystallites were rotated by 30° with respect to the mica substrate, as indicated by the white arrows. The 30°-rotated domain usually existed in the epitaxial ZnO layers grown on *c*-plane sapphire substrates and the (111) plane spinel substrates [42,43]. In contrast, the high density and uniform distribution of well-faceted ZnO crystallites preferentially occurred under oxygen ambient (Type B). Subsequently, both types of Mist-CVD-grown ZnO crystallites were used as a seed layer for the hydrothermal growth of continuous ZnO films. Figure 8c,d display the surface morphologies of the ZnO films grown on both types of ZnO seed layers through the hydrothermal process for 5 h under the same hydrothermal growth condition as that described before. Completely continuous ZnO films on either Type A or Type B seeds were evident. For the ZnO films grown on Type A seeds, they revealed a surface morphology featuring typical hexagonal mosaic structures. In addition, the 30°-rotated domains, as indicated by white arrows, were present in the ZnO layer (Figure 8c). On the other hand, the ZnO layer grown on Type B seeds exhibited a nearly flat and smooth surface morphology, as shown in Figure 8d.

XRD measurement was performed to evaluate the structural characteristic of the coalesced ZnO films grown by the seed-layer-assisted hydrothermal growth. Figure 9 shows the XRD measurements of ZnO films grown on both types of ZnO seed layers. The θ-2θ scan profiles of both samples revealed the ZnO films, featuring a *c*-axis-preferred orientation, as shown in Figure 9a. Interestingly, the obvious difference in the XRD φ-scan profiles of the two samples can be observed in Figure 9b,c. For the ZnO films grown on Type A seeds, two sets of peaks with six-fold symmetry, relative to each other through 30° rotation about the surface normal, were present in the ZnO {10$\bar{1}$1} φ-scan, indicating that two orientation variants of ZnO coexisted on the mica substrate, in agreement with

the SEM observation (Figure 8c). We deduced the in-plane crystallographic relationship between the ZnO and mica to be $[11\bar{2}0]_{ZnO} \mid\mid [10]_{mica}$ (original orientation variant) and $[11\bar{2}0]_{ZnO} \mid\mid [100]_{mica}$ (30°-rotated domain), respectively. Instead of the original orientation variant, the 30°-rotated domain was the dominant variant in the ZnO layer grown on the seed layer (Type A) under nitrogen ambient conditions. In comparison, the XRD φ-scans for ZnO $\{10\bar{1}1\}$ reflections corresponding to the ZnO film grown on Type B seeds featured merely one set of peaks with six-fold symmetry; moreover, the in-plane crystallographic relationship was confirmed to be $[11\bar{2}0]_{ZnO} \mid\mid [10]_{mica}$. The ambience during Mist-CVD growth of ZnO seeds greatly influenced the epitaxial growth behavior of ZnO crystallites on the mica substrate at the initial stage. The reason for the epitaxial orientations of the ZnO seeds on the mica substrate as a function of growth ambient is still not explicit at the moment. Nevertheless, the Mist-CVD growth environment, as well as the mica surface conditions, might influence the adatom arrangement on the mica surface, and the detailed mechanism will be clarified in future work.

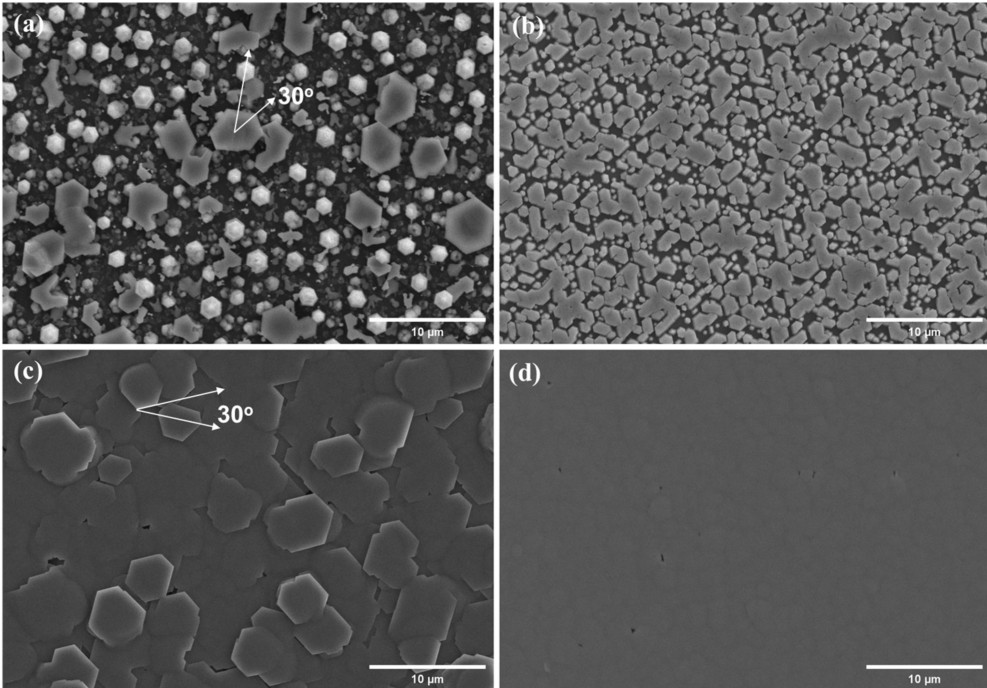

**Figure 8.** SEM images of ZnO seeds grown on as-cleaved mica substrates by Mist-CVD at 600 °C for 30 min under (**a**) nitrogen (Type A) and (**b**) oxygen (Type B) ambient. Subsequent hydrothermal growth of continuous ZnO layers on (**c**) Type A and (**d**) Type B seeds.

　　The epitaxial quality of the coalesced ZnO films grown through the seed-assisted hydrothermal growth was assessed by XRD rocking curve measurement. We performed XRC measurements in on-axis symmetric (ZnO (0002) plane) and off-axis skew-symmetric (ZnO $(10\bar{1}1)$ plane) geometries to examine the out-of-plane and in-plane mosaic spreads of the ZnO epitaxial films. Figure 10 shows the results of the XRC analysis of the ZnO (0002) and $(10\bar{1}1)$ planes for both samples. As expected, the XRC FWHM values ($\omega = 0.26°$ for (0002) plane; $\Delta\omega = 0.7°$ for $(10\bar{1}1)$ plane) of the ZnO layer grown on Type B seeds were smaller than those of the film grown on Type A seeds ($\Delta\omega = 0.41°$ for (0002) plane; $\Delta\omega = 1.6°$ for $(10\bar{1}1)$ plane). The crystal quality of the epitaxial ZnO layer grown on mica with Type B seeds was compared with those grown on the sapphire substrate under similar growth conditions [35,36].

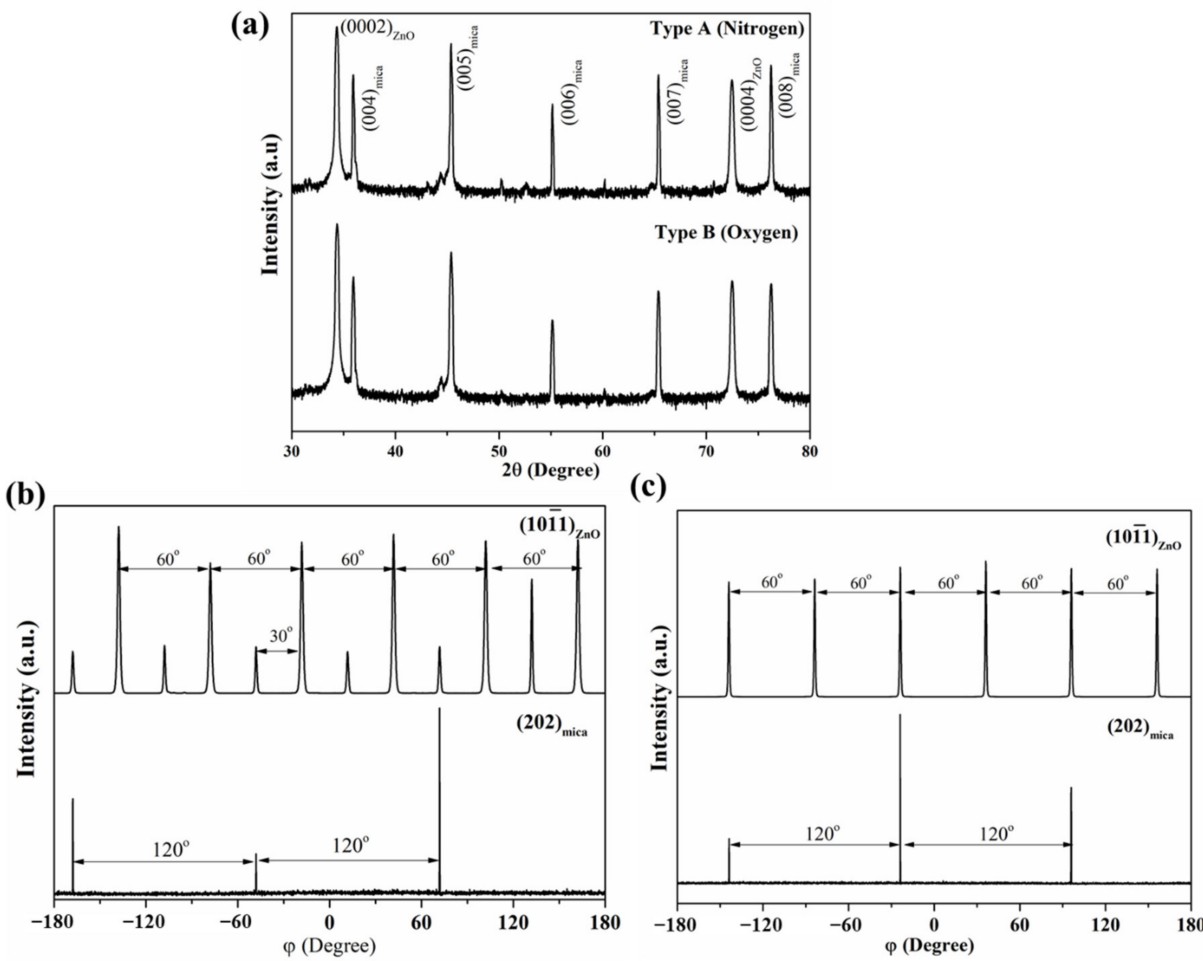

**Figure 9.** (**a**) XRD θ-2θ profiles of ZnO layers hydrothermally grown on Type A and Type B seeds. XRD φ-scan of the ZnO {10$\bar{1}$1} and mica {202} reflections for ZnO layers grown on (**b**) Type A and (**c**) Type B seeds, respectively.

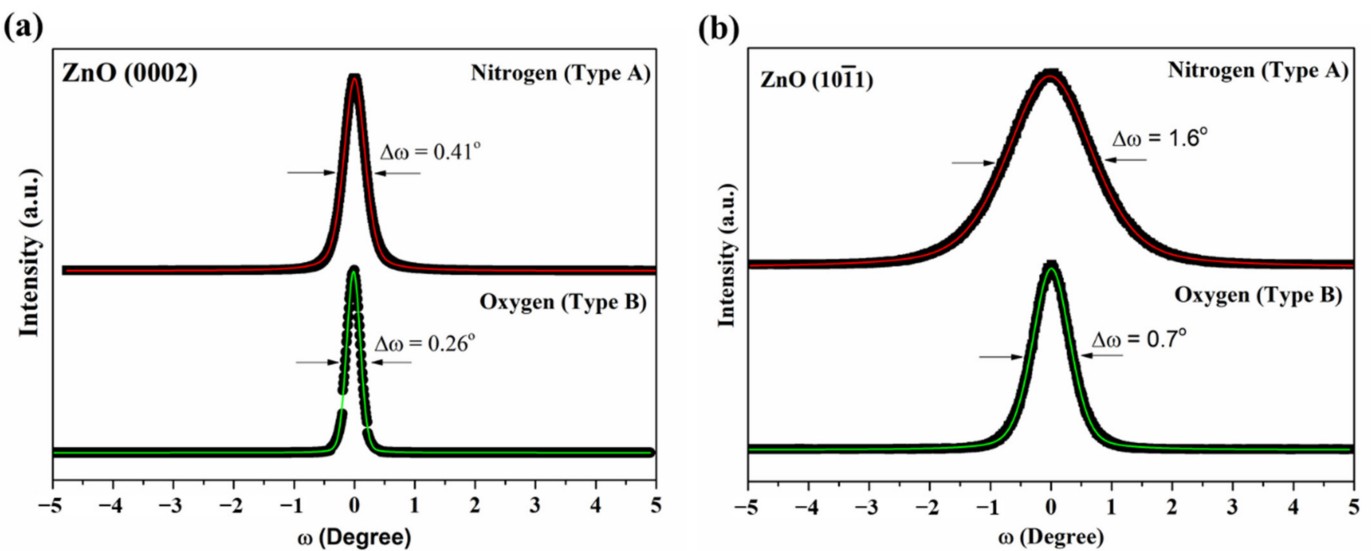

**Figure 10.** Rocking curve profiles of the (**a**) ZnO (0002) reflection and (**b**) ZnO (10$\bar{1}$1) reflection corresponding to ZnO layers hydrothermally grown on Type A and Type B seeds, respectively.

The characterization of optoelectrical and electrical properties is important for evaluating the potential application of vdW epitaxial ZnO layers in optoelectronic devices. We recorded the room-temperature photoluminescence (PL) and UV-Vis spectra of the vdW epitaxial ZnO films, corresponding to Figure 8c,d, hydrothermally grown on mica substrates with Type A and Type B seeds, respectively. In Figure 11a, the PL spectra of both samples exhibited a near-band edge (NBE) emission peak centered at 376 nm, and a broad deep-level emission (DLE) in the range of 450–700 nm, arising from defect-related emissions in the ZnO crystal. Based on the previous reports [44], this deep-level emission is attributed to the recombination of various intrinsic point defects, such as oxygen vacancies ($V_o$), zinc vacancies ($V_{Zn}$), and oxygen interstitials ($O_i$), which are usually abundant in the chemical solution-derived ZnO crystals. Figure 11b shows the UV-Vis spectra of the vdW epitaxial ZnO films grown on Type A and Type B seeds. An absorption edge near the UV region and a high transparency in the visible spectral range can be observed in both samples. The transmittance (T) of vdW epitaxial ZnO films in the visible region were observed to be in the range of 60% to 70% for Type A seeds and 75% to 85% for Type B seeds. The relatively low transmittance of the ZnO film grown on Type A seeds might be attributed to the light scattering occurring on the uneven surface of the film. The optical absorption coefficient ($\alpha$) was calculated using the following equation:

$$\alpha = -ln(T)/d \tag{1}$$

where $d$ is the thickness of the film. Figure 11c shows the corresponding $(\alpha h\nu)^2$ versus $h\nu$ (Tauc) plots. The optical band gaps of both vdW epitaxial ZnO layers were estimated to be 3.25 eV through extrapolation of the linear region of the Taue plots. Furthermore, the electrical properties of the vdW epitaxial ZnO layers grown on Type A and Type B seeds were analyzed by Hall-effect measurement, respectively. Both ZnO films showed n-type conductivity and the results of the electrical properties are summarized in Table 1. Based on the above-mentioned results, it is evident that the vdW epitaxial ZnO on the mica substrate is comparable, in terms of optical and electrical properties, with those grown on sapphire substrates through conventional solution-phase epitaxy techniques [44–46].

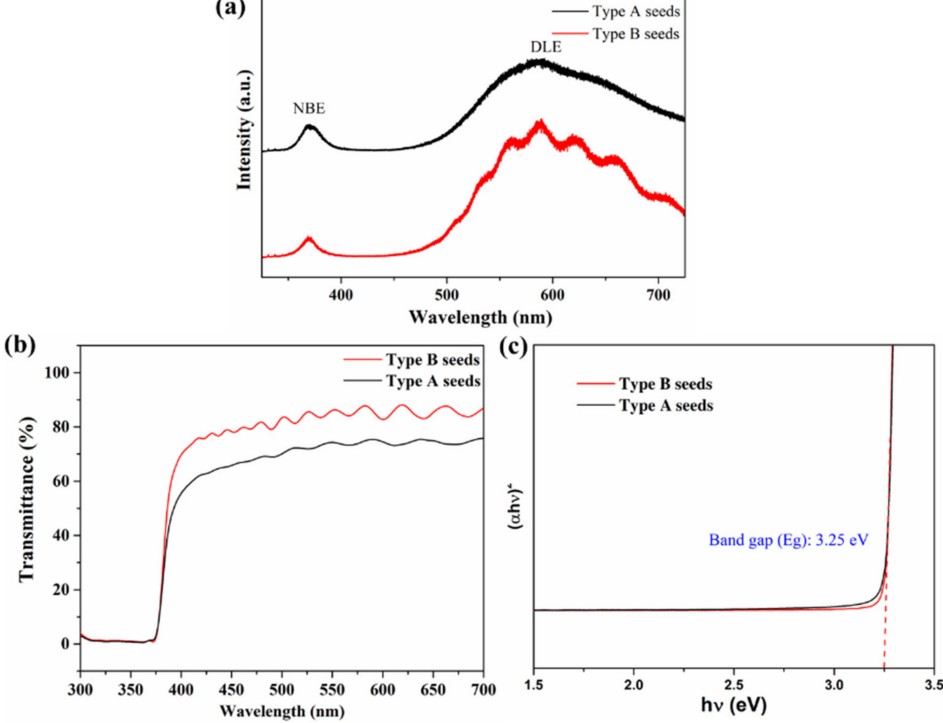

**Figure 11.** (**a**) Room-temperature PL spectra, (**b**) transmittance spectra, and (**c**) corresponding plots of $(\alpha h\nu)^2$ versus $h\nu$ of the ZnO layers hydrothermally grown on Type A and Type B seeds.

**Table 1.** Electrical properties of ZnO films hydrothermally grown on Type A and Type B seeds.

| Type of Seeds | Electrical Resistivity ($\Omega$-cm) | Carrier Concentration ($cm^{-3}$) | Hall Mobility ($cm^2V^{-1}s^{-1}$) |
|---|---|---|---|
| Type A | 0.84 | $1.5 \times 10^{18}$ | 2.92 |
| Type B | 1.05 | $6.0 \times 10^{17}$ | 11.55 |

**4. Conclusions**

In this work, we demonstrated the van der Waals epitaxial growth of ZnO films on mica substrates in a low-temperature aqueous solution. Proper annealing of the mica substrate was found to be critical for the solution-phase vdW epitaxy of ZnO micro-rods. A fully coalesced, continuous, ZnO epitaxial layer could be obtained through the addition of sodium citrate into the solution. The epitaxial relationship between the ZnO epitaxial film and mica substrate was $(0001)_{ZnO} \mid \mid (001)_{mica}$ and $[11\overline{2}0]_{ZnO} \mid \mid [10]_{mica}$, respectively. The analysis of HRTEM images revealed the near-strain relaxation of ZnO adjacent to the interface, and we, therefore, arrived at the outcome that the epitaxial paradigm of ZnO films on the mica substrate was the incommensurate vdW epitaxy. Furthermore, the seed-layer-assisted hydrothermal growth, implemented by combining the Mist-CVD process with low-temperature hydrothermal growth, improved significantly the crystal quality of the epitaxial ZnO layers on mica substrates. The in-plane alignment of Mist-CVD grown ZnO crystallites was affected by the growth ambience. The use of oxygen as a carrier gas promoted a high-density and even distribution of ZnO seeds on mica substrates, and it resulted in the growth of a ZnO epitaxial layer with a nearly flat and smooth surface morphology. Moreover, the optoelectronic and electrical properties of the vdW epitaxial ZnO films were comparable to those grown on sapphire substrates by conventional solution-based epitaxy techniques. Thus, these results led to the conclusion that the vdW epitaxial ZnO films grown on mica substrates had great potential for optoelectronic applications.

**Author Contributions:** Conceptualization, H.-G.C. and H.-S.W.; resources, H.-S.W. and S.-R.J.; writing—original draft, H.-G.C.; writing—review and editing, Y.-H.S.; methodology, T.-Y.Y. and S.-C.C.; investigation, T.-Y.Y. and S.-C.C.; formal analysis, H.-G.C., H.-S.W. and S.-R.J.; supervision, H.-G.C. All authors have read and agreed to the published version of the manuscript.

**Funding:** This research was funded by the Ministry of Science and Technology, Taiwan, R.O.C., grant numbers MOST 109-2221-E-214-018 and MOST 110-2221-E-214-004.

**Institutional Review Board Statement:** Not applicable.

**Informed Consent Statement:** Not applicable.

**Data Availability Statement:** Not applicable.

**Acknowledgments:** The authors gratefully acknowledge the use of EM000800 and XRD005100 of MOST 110-2731-M-006-001 belonging to the Core Facility Center, National Cheng Kung University, Taiwan, for HRTEM, FIB, XRD experiment, and thank the MANALAB at ISU for SEM and XRD.

**Conflicts of Interest:** The authors declare no conflict of interest.

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
