# Peer review of "Van der Waals Epitaxial Growth of ZnO Films on Mica Substrates in Low-Temperature Aqueous Solution"

_coatings, doi:10.3390/coatings12050706_

Round 1
Reviewer 1 Report
This paper is about producing micro ZnO rodes thin films. The aim of scope of the paper is suitable to the journal of the Coatings. May be, need to minor revision:
- The nano ZnO thin films can be compare to macro-ZnO films. You can read and use to this paper:
- Structural and Optical Properties of Undoped and Silver, Lithium and Cobalt-Doped ZnO Thin Films Surface Review and Letters 27 (04), 1950138, 2020
Author Response
This paper is about producing micro ZnO rodes thin films. The aim of scope of the paper is suitable to the journal of the Coatings. May be, need to minor revision:
- The nano ZnO thin films can be compare to macro-ZnO films. You can read and use to this paper:
- Structural and Optical Properties of Undoped and Silver, Lithium and Cobalt-Doped ZnO Thin Films Surface Review and Letters 27 (04), 1950138, 2020
Ans. As your suggestion, we cited this reference and discussed in the revised manuscript.
Reviewer 2 Report
The author presented a very interesting work on the importance of vdW interaction making epitaxial growth of ZnO layers on mica on mica substrates through a low-temperature hydrothermal process.
The HRTEM images are very clear and give an immediately understanding of what it was done as the same for XRD. It was a very good job. What that I will really appreciate to make complete the work and accessible to higher number of readers is the possibility to put in results an estimation of vdW interaction using Hirshfield surface calculated with Crystal Explorer (CE) program (https://crystalexplorer.scb.uwa.edu.au). It is very easy to use CE when you have generated your *.cif file with coordinates and you will better highlight vdW interaction. Please do it and cite the works correlated to Hirshfield surface of Prof. Spackmann.
Furthermore, the authors in the introduction forgot to cite relatively important work about the estimation of vdW interaction through computational work done on 2D and 3D as also the study of of 2D materials under pressure with consequence change of hybridisation and consequent affecting of chemical bonding. So, please at row 38 mention these four works:
- The Volumetric Source Function: Looking Inside van der Waals Interactions. Sci Rep 10, 7816 (2020)
- Computational and Theoretical Chemistry, 1157, 47-53 (2019)
- ACS Nano, 15(4), 6861–6871 (2021).
-Journal of computational chemistry 37 (23), 2133-2139
Author Response
Reviewer 2
The author presented a very interesting work on the importance of vdW interaction making epitaxial growth of ZnO layers on mica on mica substrates through a low-temperature hydrothermal process.
The HRTEM images are very clear and give an immediately understanding of what it was done as the same for XRD. It was a very good job. What that I will really appreciate to make complete the work and accessible to higher number of readers is the possibility to put in results an estimation of vdW interaction using Hirshfield surface calculated with Crystal Explorer (CE) program (https://crystalexplorer.scb.uwa.edu.au). It is very easy to use CE when you have generated your *.cif file with coordinates and you will better highlight vdW interaction. Please do it and cite the works correlated to Hirshfield surface of Prof. Spackmann.
Furthermore, the authors in the introduction forgot to cite relatively important work about the estimation of vdW interaction through computational work done on 2D and 3D as also the study of of 2D materials under pressure with consequence change of hybridisation and consequent affecting of chemical bonding. So, please at row 38 mention these four works:
- The Volumetric Source Function: Looking Inside van der Waals Interactions. Sci Rep 10, 7816 (2020)
- Computational and Theoretical Chemistry, 1157, 47-53 (2019)
- ACS Nano, 15(4), 6861–6871 (2021).
-Journal of computational chemistry 37 (23), 2133-2139
Ans. We appreciate that reviewer provided us the valuable recommendation and useful resource-Crystal Explorer (CE). Unfortunately, we are not very familiar with CE, and the simulation of vdW interaction using Hirshfield surface is also beyond our research field. Considering that we are not familiar with this subject, the calculated results will not be able to be rigorously and correctly verified and discussed at the current moment. Hence, we will consider to involve these results in future extension work.
Furthermore, the epitaxial growth of ZnO and other functional oxides on the mica substrates have been widely reported in the past decade. The vdW epitaxy had been well established as the major paradigm for the epitaxial growth of those oxides on mica substrates [1-5]. In this study, the interfacial structure and strain state of the ZnO film-mica substrate have been well characterized through HRTEM and GPA. In previous other works [6,7], these similar results have been also employed to verify that the epitaxial growth paradigm of ZnO crystals on mica substrates is incommensurate van der Waals epitaxy. Hence, we believe that the results of the HETEM and GPA, in this study, are sufficient to prove that the epitaxial paradigm of ZnO crystal on mica substrate is vdW epitaxy.
Finally, we thank reviewer for providing us valuable references, regarding the estimation of vdW interaction through computational work, and these will be added in the revised manuscript.
Reference:
- Chu, Y.-H. Van der waals oxide heteroepitaxy. npj Quantum Materials 2017, 2, 67.
- Ma, C.H.; Lin, J.C.; Liu, H.J.; Do, T.H.; Zhu, Y.M.; Ha, T.D.; Zhan, Q.; Juang, J.Y.; He, Q.; Arenholz, E.; et al. Van der waals epitaxy of functional moo2 film on mica for flexible electronics. Applied Physics Letters 2016, 108, 5.
- Li, C.-I.; Lin, J.-C.; Liu, H.-J.; Chu, M.-W.; Chen, H.-W.; Ma, C.-H.; Tsai, C.-Y.; Huang, H.-W.; Lin, H.-J.; Liu, H.-L.; et al. Van der waal epitaxy of flexible and transparent vo2 film on muscovite. Chemistry of Materials 2016, 28, 3914-3919.
- Bitla, Y.; Chen, C.; Lee, H.C.; Do, T.H.; Ma, C.H.; Van Qui, L.; Huang, C.W.; Wu, W.W.; Chang, L.; Chiu, P.W.; et al. Oxide heteroepitaxy for flexible optoelectronics. Acs Applied Materials & Interfaces 2016, 8, 32401-32407.
- Li, B.; Ding, L.; Gui, P.; Liu, N.; Yue, Y.; Chen, Z.; Song, Z.; Wen, J.; Lei, H.; Zhu, Z.; et al. Pulsed laser deposition assisted van der waals epitaxial large area quasi-2d zno single-crystal plates on fluorophlogopite mica. Advanced Materials Interfaces 2019, 6, 1901156.
- Utama, M.I.B.; Belarre, F.J.; Magen, C.; Peng, B.; Arbiol, J.; Xiong, Q.H. Incommensurate van der waals epitaxy of nanowire arrays: A case study with zno on muscovite mica substrates. Nano Lett. 2012, 12, 2146-2152.
- Zhu, Y.; Zhou, Y.; Utama, M.I.B.; De La Mata, M.; Zhao, Y.Y.; Zhang, Q.; Peng, B.; Magen, C.; Arbiol, J.; Xiong, Q.H. Solution phase van der waals epitaxy of zno wire arrays. Nanoscale 2013, 5, 7242-7249.

Reviewer 3 Report
- Determine the rods length of prepared samples and what are the effect of annealing temperature and concentration of sodium citrate on its, please added figure explain that behavior
- Fig (11), the ZnO layer hydrothermally grown on Type B seeds. what is the preparation condition, and why the author didn’t measure other samples to compared and determine the final behavior?
Author Response
Reviewer 3
- Determine the rods length of prepared samples and what are the effect of annealing temperature and concentration of sodium citrate on its, please added figure explain that behavior
Ans. The focus of this study is to understand how the annealing treatment of mica substrate influence the alignment of ZnO micro-rods hydrothermally grown on the mica substrates. Hence, the cross-sectional SEM observation of ZnO micro-rods/mica substrates have not been implemented yet. Due to the weak interaction between the ZnO and the mica, the resulting ZnO micro-rods/films were easily delaminated or separated from the mica substrates, when using the conventional cleaving-method for the cross-sectional specimen preparation. Thus, it was actually difficult to prepare the specimens for SEM observation of the cross-sectional ZnO micro-rods/mica substrates.
To overcome above-mentioned problems, we employed the focused ion beam (FIB) technique to prepare specimens for the cross-sectional observation. Unfortunately, limited by the project’s funding, it was not allowed to perform FIB for all of the samples shown in Figs. 2 and Figs. 3. The high fraction of ZnO micro-rods hydrothermally grown on as-cleaved or 400 oC-annealed mica substrates were oriented in random directions; in contrast, the high coverage of the epitaxial (in-plane aligned) ZnO micro-rods were exclusively grown on the 600oC-annealed mica substrate. Based on the practical application considerations, the samples of the ZnO micro-rods grown on the 600oC-annealed mica substrates were selected as the representative samples for the further cross-sectional observation. In this revised manuscript, we added the results of the cross-sectional TEM of ZnO grown on 600 oC-annealed mica substrates from a solution containing 0.25 mM and 0.5 mM sodium citrate, respectively, to demonstrate the effect of citrate on ZnO growth behavior. Based on the cross-sectional TEM images, the height of resulting ZnO micro-rods (~4.2 µm) grown from the solution with 0.25 mM sodium citrate was higher than that from the solution with 0.5 mM sodium citrate (~2.3 µm); hence, the higher concentration of sodium citrate obviously reduced the growth rate along c-axis of ZnO crystals, consistent with the results of other reported literatures. The effect of sodium citrate as a crystal habit modifier on the morphology of ZnO has been widely studied in the previous other works [1-4].
- Fig (11), the ZnO layer hydrothermally grown on Type B seeds. what is the preparation condition, and why the author didn’t measure other samples to compared and determine the final behavior?
Ans. The vdW epitaxial ZnO layer demonstrated in Figure 11 actually corresponds to Figure 8 (d). We will add a brief description regarding preparation condition in the corresponding paragraph. In the original version, we selected the sample grown under the optimal conditions (Type B seeds) as the representative sample to demonstrate the optical and electrical properties of the vdW epitaxial ZnO layer. In the revised manuscript, as your suggestion, the optical and electrical properties of the ZnO layer hydrothermally grown on Type A seeds will be also added in the revised manuscript for comparative study.
References:
- Kim, J.H.; Kim, E.M.; Andeen, D.; Thomson, D.; Denbaars, S.P.; Lange, F.F. Growth of heteroepitaxial ZnO thin films on gan-buffered Al2O3 (0001) substrates by low-temperature hydrothermal synthesis at 90 degrees C. Adv. Funct. Mater. 2007, 17, 463-471.
- Kim, J.H.; Andeen, D.; Lange, F.F. Hydrothermal growth of periodic, single-crystal ZnO microrods and microtunnels. Advanced materials 2006, 18, 2453.
- Urgessa, Z.N.; Oluwafemi, O.S.; Botha, J.R. Hydrothermal synthesis of ZnO thin films and its electrical characterization. Mater. Lett. 2012, 79, 266-269.
- Das, S.; Dutta, K.; Pramanik, A. Morphology control of ZnO with citrate: A time and concentration dependent mechanistic insight. Crystengcomm 2013, 15, 6349-6358.

Round 2
Reviewer 2 Report
----